# Cardiac Troponin I and Electrocardiographic Evaluation in Hospitalized Cats with Systemic Inflammatory Response Syndrome

**DOI:** 10.3390/vetsci10090570

**Published:** 2023-09-13

**Authors:** Michela Pugliese, Ettore Napoli, Rocky La Maestra, Mehmet Erman Or, Bengü Bilgiç, Annalisa Previti, Vito Biondi, Annamaria Passantino

**Affiliations:** 1Department of Veterinary Sciences, University of Messina, Via Umberto Palatucci, 98168 Messina, Italy; enapoli@unime.it (E.N.); rockylamaestra@gmail.com (R.L.M.); annalisa.previti1@unime.it (A.P.); vito.biondi@unime.it (V.B.); annamaria.passantino@unime.it (A.P.); 2Faculty of Veterinary Medicine, İstanbul University-Cerrahpasa, 34098 Istanbul, Turkey; ermanor@istanbul.edu.tr (M.E.O.); bengu.bilgic@iuc.edu.tr (B.B.)

**Keywords:** systemic inflammatory response syndrome, SIRS, cardiac damage, cat, cTnI

## Abstract

**Simple Summary:**

Myocardial dysfunction associated with systemic inflammatory response syndrome (SIRS) in people is identified to be related to considerably increased mortality. It has been recently also hypothesized that myocardial lesions may be present in cats with SIRS. The purpose of this study was to identify a possible myocardial dysfunction in cats with SIRS, evaluating cardiac troponin and electrocardiographic findings. Results obtained in this study demonstrate that cats with SIRS have increased cardiac troponin levels and alterations at electrocardiographic examination, consistent with the presence of myocardial dysfunction.

**Abstract:**

Several studies conducted on humans demonstrate the increase in cardiac troponins and the onset of arrhythmias in the course of systemic inflammatory response syndrome (SIRS). The aim of the current study was to assess the blood concentration of cardiac troponin I (cTnI) and electrocardiographic findings in SIRS-affected cats. Seventeen shorthair cats hospitalized with SIRS were enrolled (Group 1). SIRS diagnosis was performed based on the detection of at least two of the four criteria such as abnormal body temperature, abnormal heart rate (i.e., tachycardia or bradycardia), abnormal respiratory rate (i.e., tachypnea or bradypnea), and alterations of white blood cell number (i.e., leukocytes or band neutrophils). Ten cats screened for elective surgery such as neutering or dental procedures were evaluated as a control population (Group 2). They were considered healthy based on history, physical examination, hematological and biochemical profile, urinalysis, coprological exam, thyroxine assay, blood pressure measurement, and echocardiography. A physical examination, complete blood cell count, biochemistry test (including an electrolyte panel), electrocardiographic examination, and cTnI assay were carried out in each cat enrolled. Traumatic events, gastrointestinal, neoplastic, respiratory, and neurological disorders were identified as causes of SIRS in Group 1. In Group 1, a significantly higher concentration of cTnI than that in Group 2 was recorded (*p* = 0.004). In 37.5% of cats with SIRS, ventricular premature complexes occurring in couplets with multiform configuration were detected. Similarly, to humans, data herein reported would indicate possible cardiac damage present in cats with SIRS diagnosis.

## 1. Introduction

Systemic inflammatory response syndrome (SIRS) is a multifaceted clinical disorder featured by a dysregulated host reaction to infection that often results in a severe impaired functioning of organs or death [1,2]. The mortality recorded in humans and in veterinary medicine ranged between 40 and 70% [3,4,5]. SIRS is associated with the activation of multiple inflammatory pathways triggered by a lot of specific disorders such as trauma, infectious diseases, intoxications, or inflammation [2]. Identifying molecular and cellular mechanisms involved in organ system dysfunction may lead to developing different approaches and novel objectives for the management of this disorder.

The diagnosis of SIRS is based on the presence of clinical findings suggestive of the presence of systemic inflammation. The identification of SIRS in small animals is based on the detection of at least two of the four following criteria: including alterations in the body temperature, heart rate, respiratory rate, and white blood cell number.

Cats and dogs exhibiting two of these four clinical findings are considered affected by SIRS [2].

Often, patients affected with SIRS develop multiple organ dysfunction syndrome (MODS), defined as organ function alteration involving two or more organ systems, associated with a significant increase in mortality [6]. Identifying the underlying complications associated with SIRS and referable predisposing factors of multiorgan failure in ill patients can be favorable for decreasing mortality thanks to performing an appropriate and timely treatment [7].

Given that SIRS-affected patients have a superior risk of developing arrhythmias consequent to increased cardiac excitability, the frequent assessment of vital signs, electrocardiographic monitoring, and continuous blood pressure measurement is suitable during hospitalization [3,8,9,10]. It has been reported that organ dysfunction takes an imperative part in the determination of prognosis and survival [11]. The onset of cardiac arrhythmias and increased concentrations of cardiac troponins are correlated with worse short- and medium-term prognosis [12,13,14,15,16].

Troponin is a complex of three regulatory proteins (i.e., troponin I, troponin C, and troponin T) involved in the contractility of the cardiac and skeletal muscle apparatus [17]. Troponin I is specific for identifying the myocardial damage; indeed, its elevation is reported during injury involving cardiac myocytes [18]. Myocardial dysfunction is a common complication in sepsis setting [19]. Troponin elevation has been recommended as a biomarker to diagnose an underlying myocardial lesion during sepsis [20]; in fact, the elevation of cardiac troponins is reported in 31–80% of human patients affected by SIRS [21], following an insult to cardiac myocytes.

Moreover, in this species, it is consistently believed that cTnI and electrocardiographic findings may play a significant function in SIRS management [5].

The goal of this study was to assess the possible cardiac involvement in cats hospitalized with SIRS diagnosis through electrocardiographic examination and the evaluation of cTnI (cardiac troponin I) blood concentration.

## 2. Materials and Methods

### 2.1. Animals

Seventeen shorthair cats referred in the emergency room of the Veterinary Teaching Hospital of the University of Messina, with SIRS diagnosis, were retrospectively included (Group 1). The identification of SIRS was performed in the presence of at least two of the four following parameters, recorded during the course of the clinical examination at the time of admission in emergency: abnormal body temperature (≤37.8 or ≥39.7 °C); abnormal heart rate (tachycardia ≥ 225 beats/min or bradycardia ≤ 140 beats/min); abnormal breath rate (tachypnea ≥ 40 or bradypnea ≤ 20 breaths/min); and white blood cell (WBC) abnormalities (WBC ≥ 19,500 or ≤5000 k/μL or band neutrophils ≥ 5%) [22].

Cats weighing <2 kg, aged <6 months or >12 years, or with a history of cardiac or evidence of kidney diseases [23,24,25] were not included.

Ten adult domestic shorthair cats screened for elective surgery such as neutering or dental procedures, considered healthy, based on history, physical examination, hematological and biochemical test, urinalysis, coprological exam, thyroxine assay, blood pressure measurement, and echocardiography were enrolled in this study as the control group (Group 2).

### 2.2. Ethics Statement

This study was carried out following the standards included in the Guide for the Care and Use of Laboratory Animals and Directive 2010/63/EU. The procedure and methods employed were ethically examined and approved by the Ethics Committee of the Department of Veterinary Sciences of Messina University (No. 10/2019). All owners were informed of the purpose of this study. A written informed consent before the inclusion of their cats in this study was signed.

### 2.3. Clinical Score

Each animal underwent a physical examination, and a scoring system to categorize the clinical parameters identifying the presence of SIRS was carried out; each variable was scored from 0 to 1 in accordance with the absence or presence of the clinical finding detected (Table 1).

Systemic blood pressure was measured using a non-invasive high-definition oscillometric device (VET HDO; S and B MedVet, Babenhausen, Germany) within 24 h after admission [26]. Conforming to the manufacturer’s instructions, the cuff was positioned on the tail. The occlusion of the coccygeal artery was obtained by positioning the cuff on the base of the tail, with the artery arrow disposed along the ventral midline. A minimum of five measurements were recorded. If the oscillometric device did not provide a complete set of blood-pressure-assessed values (values of systolic, diastolic, and mean arterial blood pressure) for three consecutive measurements, the cuff was repositioned and measurements repeated. The mean arterial blood pressure (MAP) was considered as a minimum of five measurements showing systolic blood pressure measurements that diverged <20% from each other.

### 2.4. Collection of Blood Samples and Laboratory Procedures

Blood samples were collected from the cephalic or jugular vein into K_3_EDTA and serum within 24 h of admission. The tube containing K_3_EDTA was used to carry out a complete cell blood count and to assess plasmatic cTnI levels. A tube provided with a cloth activator was centrifuged at 2000× *g* for ten minutes to obtain the serum, which was used for biochemistry including electrolyte evaluations. Biochemistry on collected sera was performed also for the evaluation of alanine-aminotransferase, glucose, urea, total protein, albumin, creatinine, sodium, potassium, chlorine, and total Ca^2+^. cTnI levels were analyzed using commercially available high-sensitivity immunoassay (ADVIA Centaur CP TnI-ultra; Siemens Healthcare Diagnostics, Erlangen, Germany) [27]. To evaluate feline white blood cell morphology, blood smears stained with May-Grunwald Giemsa (Aerospray Wescor, Delcon, Milan, Italy) were microscopically examined.

### 2.5. Electrocardiographic Examination

Cats were manually restrained and placed in right lateral recumbency. Front legs were held perpendicularly positioned to the long axis of the body, while hind limbs were held in a neutral semi-flexed position [28]. Electrodes were positioned on the skin using flattened alligator clips near the olecranon on the caudal facet of the forelimb and over the patellar ligaments on the cranial facet of the hind limbs [29]. Isopropyl alcohol was used to ensure optimal electrical interaction between electrodes and the skin [29]. Electrocardiogram traces for standard 6-lead (leads I, II, III, aVR, aVL, and aVF) were recorded for 2 min. The paper speed was set at 50 mm/s. Paper sensitivity was set at 20 mm/mV. Evaluation of ECGs was performed according to the standard methods with a caliper and a ruler [30]. ECG parameters considered included cardiac rhythm (i.e., normal sinus rhythm, sinus arrhythmia, and pathological arrhythmias), heart rate (HR), P wave amplitude and duration, PQ interval duration, amplitude and duration of the QRS complex, ST-segment elevation or depression, T wave amplitude, and QT interval (QT) duration [30]. In addition, the corrected QT interval (QTc) according to the logarithmic formula (QTc = log600 × QT/logRR) was determined [31]. The heart rate expressed in bpm was determined by the number of QRS complexes in a 3 s interval, multiplying this value by 20. The PR interval was determined from the onset of the P wave to the start of the QRS complex. The QT interval was calculated from the beginning of the QRS complex to the end of the T wave. A mean of 5 cardiac beats was expressed as the result for each variable [30]. Cardiac rhythm was also calculated evaluating twenty beats on lead II. Analysis of feline P-QRS-T deflections was determined [32]. The equation considering the mean electrical axis = arctan (Iamp, aVFamp) × 180/π was applied to establish the mean electrical axis (MEA) [30]. For rhythm analysis, specific criteria were applied [32]. Normal sinus rhythm was defined as sinus rhythm associated with a normal HR (120–220 bpm) and with less than 10% variation in R-R intervals. In the presence of a sinus rhythm with a normal HR (120–220 bpm) and more than 10% variation in R-R intervals, a sinus arrhythmia was identified. In the presence of three consecutive sinus complexes associated with an HR exceeding the superior normal limit, sinus tachycardia was detected, while sinus bradycardia was identified as more than three consecutive sinus complexes associated with an HR under the lower normal limit. Premature narrow QRS complexes with a P wave presenting an abnormal morphology were assessed as atrial premature complexes. Supraventricular tachycardia was identified for the presence of more than three consecutive atrial premature complexes at an HR above the upper normal limit. Ventricular premature complexes (VPCs) were defined by the presence of wide and bizarre QRS complexes in the absence of an associated P wave. The accelerated idioventricular rhythm was detected for the detection of more than three VPCs at a normal HR. Ventricular tachycardia was identified as more than three VPCs at an HR above the upper normal limit. The second-degree atrioventricular block was detected as one or more P waves not followed by a QRS complex, while others were conducted with an associated QRS complex. The third-degree atrioventricular block was diagnosed detecting complete dissociation between atria and ventricles, with an atrial rate not correlated with the ventricular rate.

### 2.6. Statistical Analysis

Statistical analyses were carried out applying appropriate software (GraphPad Prism 8, version 8.4.0, Graph Pad Software, San Diego, CA, USA). The normal distribution of data was assessed by visual inspection of graphs and by the Shapiro–Wilk test. Quantitative variables normally distributed were expressed as mean ± standard deviation (SD). The median and interquartile range (IQR) were used to present continuous variables non-normally distributed (cTnI). The *t*-student test was used to compare parametric data, while the Mann–Whitney U-test was applied for non-normally distributed non-parametric data. *p*-value was set at ≤0.05.

## 3. Results

Group 1 was composed of 17 domestic shorthair cats (10 males and 7 females, aged 1.3–11.2 years (median age 6.2 years, IQR 8.8 years) and weighing 2.2–6.1 kg (average 3.75)). All cats of Group 1 presented two or more criteria suggestive of SIRS; 6/17 cats showed 2/4 of the SIRS criteria, while 11/17 showed ≥3/4. The causes that led to SIRS were traumatic in 8/17 (47.05%) and gastrointestinal in 5/17 (29.41%); neoplasia was detected in 2/17 (11.76%), respiratory in 1/17 (5.88%), and neurological in 1/17 (5.88%). The mean clinical score of Group 1 was significantly higher (3.06 ± 3.9) than that of Group 2 (1 ± 0.8) (*p* < 0.001). Table 2 summarizes the results regarding body temperature, heart rate (bpm), respiratory rate (brpm), and WBC of both groups.

No significant differences between the two groups were detected regarding body temperature and WBC count; conversely, significant differences in heart rate (*p* = 0.001) and respiratory rate (*p* = 0.005) were recorded.

Group G2 was composed of 10 healthy cats (7 males and 3 females), aged 1–8 years (median age 6 years, IQR 3.25 years) and weighing 2.0–5.2 Kg (average 3.45).

Body temperature was altered in ten cats of Group 1 (i.e., 58.82%). Eight cats belonging in Group 1 showed alterations in heart rate; tachycardia was observed in 4/17 (23.52%) and bradycardia in 4/17 (23.52%). In five patients of Group 1 (29.41%), an increase in the respiratory rate was recorded. All patients belonging to Group 1 showed alteration in the number of leukocytes; in particular, leukocytosis was recorded in 64.70% of cats (11/17), while leukopenia was detected in 6 patients (i.e., 35.29%). None of the cats presented ≥5% band neutrophils. Regarding Group 2, an increase in body temperature was detected in two cats (i.e., 20%). Only one cat (20%) showed tachycardia, while no alterations in the respiratory rate were detected. Leukocytosis was recorded in 30% of cats (3/10).

Values of MAP were lower in cats of Group 1 (91.12 ± 19.5) than in those in Group 2 (112.20 ± 16.4) (*p* = 0.02).

Thirteen cats of Group 1 (76.47%) showed cTnI values outside of normal ranges. The mean value of cTnI in Group 1 was 0.68 ng/mL (median 0.48; IQR 0.40–0.99), which was significantly higher (*p* = 0.004) compared with that of Group 2 (0.045; IQR 0.011–0.037) (Table 2). Mean arterial pressure values were significantly higher in Group 2 (130.3 ± 11.6) than in Group 1 (104 ± 8.4) (*p* = 0.036).

Although all means of electrocardiographic variables were in the physiological ranges, the heart rate was significantly lower in Group 1 than in Group 2 (*p* = 0.034), as well as R amplitude (*p* = 0.048). PR, QRS, and QT intervals were significantly longer (Table 3) in Group 1 than in Group 2 (*p* = 0.024, *p* = 0.043, and *p* = 0.027, respectively).

In 5 (29.41%) cats belonging to Group 1, the presence of ventricular premature complexes was observed. Ventricular premature complexes occurred in couplets with multiform configuration. ECG variables expressed as quantitative data are summarized in Table 3.

## 4. Discussion

SIRS is a severe clinical condition connected with considerable morbidity and mortality in human and veterinary medicine. Although progress in the approach to the treatment of SIRS during the last years has determined a decrease in the morbidity and mortality rates in human patients, the burden continues to be profound for the management in dogs and cats causing a longer duration of hospitalization, higher treatment costs, and higher mortality rate, compared with other critical disorders. The main objective in the management of a patient affected with SIRS is to perform an early diagnosis and, consequently, an adequate treatment [33]. A prompt diagnosis of SIRS focuses on detecting symptomatic patients as early as possible, so they have the best chance for successful and effective therapy and to better prognosis [34]. SIRS continues to be the focus of several studies in which the interpretation of mechanisms involved in the pathophysiology is continually reconsidered [32,33,34,35,36,37], and the significance of species-specific dissimilarities is being documented in the literature [37]. Despite the elevated mortality rate in both dogs and cats affected with SIRS, the available literature is limited, especially concerning cats.

Fever, tachycardia, tachypnea, and an abnormal white blood cell count are described in the literature as criteria applying in cats to diagnose SIRS [22], but they seem to have a lack of specificity and do not provide information regarding the etiology of the disease, often determining a misclassification of sick patients in clinical settings.

In accordance with the proposed guidelines for diagnosing SIRS in cats [5], the patients enrolled in the current study showed the presence of three of the four SIRS criteria. Cats affected with SIRS showed significantly lower heart rates than the control group. It is important to highlight that clinical findings of SIRS diverge in dogs and cats. As reported in a previous study, cats affected with SIRS present an unexpected response involving the heart rate more than other species that reported a prevalence. About 66% of cats affected with SIRS showed a low heart rate [5]. Indeed, cats frequently do *not* mature a hyperdynamic response (e.g., no red mucous membranes) and are more likely to present relative bradycardia and hypothermia. The mechanisms involved are unknown, appearing as unique in the feline species [5].

The main objective of the current study was to evaluate the presence of myocardial dysfunction in SIRS cats. The presence of cellular injury is confirmed by the release in the circulation of biomarkers from the cell.

Biomarkers are usually used in clinical practice as efficient diagnostic tools in the determination of staging, categorizing, and primary treatment selection; and subsequently to diagnosis for monitoring the treatment, performing a supplementary treatment, or monitoring recurrent diseases [38]. Biomarkers are released by organs in response to various insults. They may also be applied before diagnosis as screening and risk evaluation [39].

In relation to SIRS, the application of diagnostic and prognostic biomarkers may be a potentially important application in overly sensitive yet poorly specific SIRS score criteria and represents an auspicious research field for veterinary disease. Myocardial injury is common in critically human patients [40] and dogs [41,42].

Cardiac troponin I (cTnI) is the most valuable and specific biomarker for detecting cardiomyocyte injury [40]. The cTnI assay is applied in human intensive care units as a prognostic marker of myocardial injury, both in human patients with structural cardiac disease [39,40] and in human patients with noncardiac disorder that secondarily involves the heart [15]. In veterinary medicine, a high serum concentration of cTnI is reported as a negative prognostic marker of death in dogs affected with SIRS [42,43,44]. Significantly higher levels of cTnI have been assayed also in cats with hypertrophic cardiomyopathy [45,46,47] and no specific primary cardiac diseases such as hyperthyroidism [48], renal disease [23], critical illness [48,49], and hypertension [50]. This study showed that an increase in cTnI suggesting the presence of myocardial injury is common in cats affected with SIRS, similarly to dogs [51]. While the pathogenesis of myocardial injury can vary and be influenced by underlying disease, cTnI seems to be a considerable marker of myocardial injury in SIRS settings [51,52]. Indeed, in human medicine, cardiac injury, better known as “myocardial hibernation”, has been reported in critically human patients and in dogs with sepsis experimentally induced [16,53,54]. It is considered a protective mechanism of the body during systemic inflammation characterized by increased end-diastolic and end-systolic ventricular volumes, and systolic and diastolic ventricular dysfunction, with a negative impact on survival [16,53,54]. It is associated with an increase in blood levels of cTnI and with an unfavorable prognosis in patients affected with SIRS [16,53,54]. Some studies have reported a possible correlation concentration of cTn and the degree of cardiac dysfunction and plasma concentrations [16]. They are limited knowledge about myocardial hibernation in dogs naturally affected with SIRS, although it is likely to occur [55,56].

During SIRS, there is an elevated cardiac metabolic need [57], strongly correlated with an increased request in cardiac blood flow. Patients with pre-existing subclinical cardiac dysfunction may have a mismatch, which can lead to ischemia [58]. The lack of the number of cats for each cause of SIRS has not allowed for the evaluation of the possible intragroup correlation based on different etiology. Another theory, based on the lack of evidence of myofibril ischemic damage in the setting of SIRS, hypothesizes that during cardiac stress, the cytosolic cTnI fraction can be released without injury to the cell’s structural apparatus, but only injury to its membrane [51]. It has also been theorized that the increase in cardiac troponin may be associated with bacterial myocarditis [21] determining a release of troponins. Indeed, in the setting of Gram-negative bacteremia and sepsis, the onset of myocardial depression and ventricular dilatation following an outflow of cytokines (IL1*β*, IL-6, and TNF*α*), nitric oxide, and endotoxins has been reported. Other low-significant theories such as activation of free and superoxide radicals are reported in experimental studies involving rats [59]. Ventricular wall stress-mediated is likewise postulated in human settings [60,61].

Ventricular premature complexes were arrhythmias detected in cats affected with SIRS. They are ectopic beats that develop from within the ventricles. They are common and happen in a broad variety of clinical situations and can present in people both with and without cardiac disease. In human medicine, the causes of arrhythmias associated with SIRS are still debated [62,63,64], although several studies have documented the presence of sinus tachycardia [63], atrial fibrillation [64], low QRS complex [5], and prolongation of the QT interval [9] in critically ill patients. It has been postulated that when SIRS arises, distress to the cardiovascular system induces an altered myocardial excitability related to a reduced supply of oxygen to the myocardium [62]. Furthermore, altered activation of the sympathetic nervous system may occur during sepsis, which can play an important role in the genesis of tachyarrhythmia [62] or the low heart rate documented in affected cats.

This is the first study that documents ECG recording, laboratory, and clinical findings in cats with SIRS. The results of this study confirm the hypothesis that feline SIRS is associated with a potential myocardial injury.

Although in the population studied, other potential causes of cTnI increase cannot be excluded, the cTnI value should be considered as a sensitive biomarker of cardiac involvement in cats with SIRS, as well as in dogs [43,44] and humans [15,21,40]. Given that these guidelines were recognized based on cats with sepsis, it is reasonable to infer that the increase in cTnI recorded in the present study suggests a possible cardiac involvement during SIRS. Data herein reported suggest the possibility of using cTnI as a parameter in SIRS clinical settings to improve the formulation of a reasonable and appropriate treatment plan for patients.

## 5. Conclusions

Results of the current study describe the ECG changes and cTnI values in cats affected with SIRS suggesting a possible involvement of the heart. The addition of cTnI assay cardiac troponins in the feline SIRS protocol may support risk-stratify patients so that echocardiographic examination and cardiac consultations can be performed at the onset of symptoms. A principal limitation of this study is the lack of echocardiographic examination to define a pre-existing subclinical cardiac disease. However, this study reports pilot data, and additional studies enrolling a larger patient population are indispensable to delineate the role of cardiac testing in the setting of feline SIRS.

## Figures and Tables

**Table 1 vetsci-10-00570-t001:** The scoring system applied during clinical evaluations of cats affected by SIRS.

Variables	0 (Normal)	1 (Clinical Parameters of SIRS)
Body temperature (°C)	≤37.8 or ≥39.7	≤37.7 or >39.7
Heart rate (bpm) *	≤140	≥225
Respiratory rate (brpm)	≥20 or ≤40	≤20 or ≥40
White blood cells (k/μL)	≥5000 or ≤19,500 or band neutrophils ≤ 5%	≥19,500 or ≤5000 or band neutrophils ≥ 5%

* bpm = beats per minute; brpm = breaths per minute.

**Table 2 vetsci-10-00570-t002:** Mean, standard deviation of clinical variables, clinical score, and cTnI values in SIRS (Group 1) and healthy cats (Group 2).

Clinical Variables	Group 1	Group 2
Temperature (°C)	35.8 ± 8.3	38.6 ± 0.5
Heart rate (bpm)	176 ± 59.5 *	139.6 ± 10 *
Respiratory rate (brpm)	35.2 ± 10.6 *	19.5 ± 3.61 *
White blood cells (k/μL)	18.10 ± 12.7	11.8 ± 3.9
cTnI (ng/mL)	0.68 ± 1.01 *	0.045 ± 0.48 *

* Statistical significance between columns (*p* < 0.05).

**Table 3 vetsci-10-00570-t003:** Electrocardiographic variables of cats with SIRS (Group 1) and healthy cats (Group 2). Data are expressed as the mean and standard deviation.

Electrocardiographic Variables	Group 1	Group 2
Heart rate (bpm)	116 ± 59.4 *	144 ± 27.5 *
MEA (degree)	49.4 ± 62.3	29.4 ± 11
P duration (msec)	36 ± 8.4	34.2 ± 0.12
P amplitude (mV)	0.14 ± 0.05	0.15 ± 0.05
PR duration (msec)	66.6 ± 10 *	48 ± 10 *
QRS duration (msec)	46 ± 13.4 *	37 ± 2.6 *
R amplitude (mV)	0.35 ± 0.26 *	0.67 ± 0.21 *
ST deviation (mV)	0.01 ± 0.01	0
T amplitude (mV)	0.15 ± 0.13	0.2 ± 0.1
QT duration (msec)	206 ± 77.7 *	155 ± 34 *
QT duration corrected (msec)	285 ± 66.1 *	207 ± 25 *

* Statistical significance between columns (*p* < 0.05); MEA = mean electrical axis.

## Data Availability

Data may be found by contacting the corresponding author (michela.pugliese@unime.it).

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
