# Peer review of "Cardiac Troponin I and Electrocardiographic Evaluation in Hospitalized Cats with Systemic Inflammatory Response Syndrome"

_vetsci, 2023, doi:10.3390/vetsci10090570_

Round 1
Reviewer 1 Report
This is an interesting study describing for the first time ECG and cTnI changes in cats affected with SIRS. The study is apparently well designed but presents several constraints in the described methodology, and results presentation. Also the manuscript as a whole requires an intensive revision of the English language, as well as the referencing in the text, as many references are exchanged and do not correspond to the subject matter.
Line 33 – Keywords, please add cTnI
Line 53 - Reference (3) is exchanged and does not correspond to the subject matter.
Reference 8 is missing in the text
Line 55 Please change “electrocardiography monitoring” by “electrocardiographic monitoring”
Line 83/4 - “Bradycardia” and “tachycardia” refer to heart rate and not pulse rate, please revise the whole manuscript including the abstract.
Also pulse rate and heart rate cannot be used interchangeably because they aren´t the same and not always coincide. SIRS diagnose criteria refers to heart rate.
Line 83 – How was pulse rate taken?
Line 86 – Please use “band neutrophils” instead of “banded neutrophils”, please revise the whole manuscript including the abstract
Line 88 – “Ecocardiography” instead of “echo”
Lines 90/3 – What were the reasons for animals in group 2 to be blood sampled? Was it on purpose for this study or were there other reasons? How were this animals enrolled?
Table 1 – Please use “band neutrophils” instead of “banded neutrophils”
Lines 109/14 – were these measurements always made by the same personnel? What was the animal´s position? As it always the same?
Line 117 – What was the use of the heparin tube?
Line 118 – What was the methodology used to identify the band neutrophils?
Line 119 - Please express centrifugation force in”G”
line 120 – the ionogram is a part of the biochemistry study…
Line 121 – What transferase did you study ALT or AST?- please use the term “glucose” instead of “total glucose”
Lines 120/2 - Please consider to remove information on biochemistry laboratory procedures. These data is not presented or discussed on results and discussion sections, nor seems to add relevant information on the subject.
Line 123 – Authors should state the brand and manufacturer of the cTnI immunoassay. The bibliographic citation used (30) does not correspond.
Lines 182/ - Please consider to move the sentence “Results regarding body temperature, pulse rate (bpm), respiratory rate (brpm), and WBC of both groups are summarized in Table 2.” To the beginning of the sentence “No significant differences between groups were observed in terms of body temperature and WBC count...”
Line 198 – Did any animal presented band neutrophils?
212 – Table 3 Once again author´s need to clarify what and how they measured, heart or pulse rate?
What does MEA means? All the abbreviations used in the table should be defined
Lines 230/8 – Consider to remove. This information is beyond the scope of this study
Lines 278/80 – Consider to remove the SIRS diagnose criteria, as they are well defined previously
Line 299 – This study is performed on cats!
References 47 and 49 are exactly the same
the manuscript as a whole requires an intensive revision of the English language
Author Response
Dear Reviewer,
Thank you very much for your time and all your comments.
We thank you for your precise and thoughtful comments and constructive criticism, which has led to a better manuscript.
We revised the manuscript concerning the suggestions and more detailed answers are given below.
The changes made in the manuscript to address comments are marked up using the
“Track Changes” function.
- This is an interesting study describing for the first time ECG and cTnI changes in cats affected with SIRS. The study is apparently well designed but presents several constraints in the described methodology, and results presentation. Also the manuscript as a whole requires an intensive revision of the English language, as well as the referencing in the text, as many references are exchanged and do not correspond to the subject matter.
- Thank you for your comments. A revision of the English language, as well as the referencing in the text, has been performed.
- Line 33 – Keywords, please add cTnI.
- Done.
- Line 53 - Reference (3) is exchanged and does not correspond to the subject matter.
Reference 8 is missing in the text.
- References have been corrected in the whole manuscript.
- Line 55 Please change “electrocardiography monitoring” by “electrocardiographic monitoring”
- Done.
- Line 83/4 - “Bradycardia” and “tachycardia” refer to heart rate and not pulse rate, please revise the whole manuscript including the abstract. Also pulse rate and heart rate cannot be used interchangeably because they aren´t the same and not always coincide. SIRS diagnose criteria refers to heart rate.
- “Bradycardia” and “tachycardia” has been revised in the whole manuscript including the abstract and corrected terms have been applied.
- Line 83 – How was pulse rate taken?
- Sorry, it is considered heart rate.
- Line 86 – Please use “band neutrophils” instead of “banded neutrophils”, please revise the whole manuscript including the abstract.
- Band neutrophils” instead of “banded neutrophils” have been revised in the whole manuscript including the abstract
- Line 88 – “Ecocardiography” instead of “echo”
- Done.
- Lines 90/3 – What were the reasons for animals in group 2 to be blood sampled? Was it on purpose for this study or were there other reasons? How were this animal enrolled?
- Information required have been included.
- Table 1 – Please use “band neutrophils” instead of “banded neutrophils”.
- Done.
- Lines 109/14 – were these measurements always made by the same personnel? What was the animal´s position? As it always the same?
- Information required have been included. Measurement were made by two Authors (M.P., R.L.M.).
- Line 117 – What was the use of the heparin tube?
- The sentence has been corrected.
- Line 118 – What was the methodology used to identify the band neutrophils?
- Information required have been included.
- Line 119 - Please express centrifugation force in”G”
- Done.
- line 120 – the ionogram is a part of the biochemistry study….
- Of course. We have preferred to specify, according the suggestion of another reviewer.
- Line 121 – What transferase did you study ALT or AST?- please use the term “glucose” instead of “total glucose”.
- Done
- Lines 120/2 - Please consider to remove information on biochemistry laboratory procedures. These data is not presented or discussed on results and discussion sections, nor seems to add relevant information on the subject.
- Thank you for the suggestion, but another reviewer has required that.
- Line 123 – Authors should state the brand and manufacturer of the cTnI immunoassay. The bibliographic citation used (30) does not correspond.
- The reference has been corrected.
- Lines 182/ - Please consider to move the sentence “Results regarding body temperature, pulse rate (bpm), respiratory rate (brpm), and WBC of both groups are summarized in Table 2.” To the beginning of the sentence “No significant differences between groups were observed in terms of body temperature and WBC count...”
- Done.
- Line 198 – Did any animal presented band neutrophils?
- Information required have been included.
- 212 – Table 3 Once again author´s need to clarify what and how they measured, heart or pulse rate?
What does MEA means? All the abbreviations used in the table should be defined.
- Information required have been included.
- Lines 230/8 – Consider to remove. This information is beyond the scope of this study.\
- Lines 278/80 – Consider to remove the SIRS diagnose criteria, as they are well defined previously.
- Line 299 – This study is performed on cats!
- Sorry for the type!
- References 47 and 49 are exactly the same.
- The reference has been corrected.
Reviewer 2 Report
Version 2 that I downloaded was identical to version 1. I cannot obtain the original version to compare edits to.
The English is improved but still requires improvement.
Author Response
Dear Reviewer,
Thank you very much for your time and all your comments, but I believe there is a mistake. The manuscript has been strongly revised and improved.
We revised the manuscript concerning the suggestions of other reviewers. The changes made in the manuscript to address comments are marked up using th
“Track Changes” function.
Reviewer 3 Report
Cardiac Troponin I and Electrocardiographic Evaluation in Hospitalized Cats with Systemic Inflammatory Response Syndrome
This is a very interesting study and this information may be helpful, however, there are many components of the manuscript and study design that need to be clarified.
Simple Summary
Line 13 – recommend rewording “Myocardial dysfunction is associated with systemic inflammatory response syndrome (SIRS) in people.”
Line 13 – it would be helpful to have a sentence on why it matters – does it affect outcome?
Line 15 – recommend reworking the purpose – “the purpose of this study was to evaluate myocardial dysfunction in cats with SIRS via cardiac troponin and ECG findings.” Or similar
Line 17 – the results of this study demonstrate cats with SIRS have increased cardiac troponin levels and ECG alterations, consistent with myocardial dysfunction.
Abstract
Line 19 – recommend “human studies have demonstrated…” Again, does it matter?
Line 20 – recommend rewording – this study evaluated the concentration of Troponin I and ECG findings…
Line 23 – while 10 healthy cats were evaluated as a control population
Line 27 – replace “breath” with respiratory
Overall it would be more helpful to separate how cats were included – criteria met and then talk about what diagnostics were performed in all cats once they were enrolled – ie: troponin and ECG
Define healthy cats? How were they enrolled?
More information about ECG in both groups should be provided
Introduction
Line 37 – citations all at the end of the sentence
Line 40 – Indent for new paragraph
Line 42-44 – this sentence doesn’t seem to add much to the point of the intro
Line 46 – “prevention” should likely not be in this sentence since it’s not the point of the study
Line 47 – “presence of at least 4 criteria” reads confusing because they don’t need to meet all criteria
Line 50 – cats and dogs need to meet a different # of criteria, so this should be reworded
Line 51 “numerous studies...” this is not relevant to the intro and should not be here
Line 54 – this should be reordered – discuss the effects of SIRS on the body (MODS) and then more specifically on cardiac effects and prognosis
Line 61 – this paragraph needs to be reorganized/rewritten; consider “Troponin I is specific for myocardial damage and elevation of troponin I is reported in…”
Line 68 – remove this sentence
Line 69 – citation?
Line 69- I am not sure this is true; “consistently supposing believed” is grammatically incorrect
Line 72 – reword “the goal of this study”
Line 75 – remove this sentence
Materials/Methods
Line 82 – during hospitalization should not be criteria – what time frame? What time frame was it diagnosed with data collected
Line 87 – why were these excluded? Based on what?
Line 87 – history of cardiac disease based on what? History of kidney disease based on what?
Line 88 – this is unclear and makes it seem biased – not all cats had an echo and some were excluded based on the echo?
Line 90 – how were they recruited? Did they also need to weight > 2kg and between 6 months to 12 years?
Line 103 – at what time frame?
Line 125 – what time frame?
Line 133-163 – too much information
Line 133 – evaluated by who? A cardiologist?
Results
Line 176 – what does this mean?
Line 177 – this should not be at the beginning of this paragraph
Line 182 – information on which criteria they met should be provided – each cat needed to meet 2/4 – which 2 did they meet
Line 192 – doesn’t seem like information that needs to be presented; they have to have significantly different scores in order to meet inclusion criteria
Line 195 – is this supposed to be bradycardia?
Line 194 – confusing statement; state how many were tachycardic and how many were bradycardic
Line 195 – needs grammatical correction
Line 192-202 – information should be provided for both groups and statistical difference provided for all values; not just MAP and troponin
No information was even provided about BP (in text and table)
Line 204 – HR is good to note; do other values matter? How often are these really looked at?
Line 210 – what about group 2?
Line 210 – what does this mean?
What cats had echos? Why did they have echos? It is mentioned earlier that some cats had echos and were excluded but there is no mention in the results section.
Discussion
Line 217 – what do you mean significant?
Line 218 – “following…” should be removed
Line 219 – doesn’t seem to contribute to the discussion
Line 221 – this sentence is not grammatically correct; also not necessarily true – diagnosis is the first important step
Line 222 – need grammatical corrections
Line 224 – is this relevant to the study?
Line 230 – run on sentence; needs to be adjusted
Line 230 – this is a disjointed paragraph
Line 235 – troponin is not being evaluated as a biomarkers for SIRS
Line 235-240 – this doesn’t make sense because this study is not looking at biomarkers for diagnosing SIRS
Line 246 – involve, not involves
Line 249 – it’s unclear what the point of these statements are
The discussion is very unorganized and hard to follow – based on this discussion it is hard to know what the points are being made
Unsure the conclusions and how they are aid in diagnosis or management
Author Response
Dear Reviewer,
Thank you very much for your time and all your comments.
We thank you for your precise and thoughtful comments and constructive criticism, which has led to a better manuscript.
We revised the manuscript concerning the suggestions and more detailed answers are given below.
The changes made in the manuscript to address comments are marked up using the
“Track Changes” function.
- This is a very interesting study and this information may be helpful, however, there are many components of the manuscript and study design that need to be clarified.
- Thank you for your comments. The whole manuscript has been revised and improved, in accordance with the suggestion given.
Simple Summary
- Line 13 – recommend rewording “Myocardial dysfunction is associated with systemic inflammatory response syndrome (SIRS) in people.”
- Done.
- Line 13 – it would be helpful to have a sentence on why it matters – does it affect outcome?
- Done.
Line 15 – recommend reworking the purpose – “the purpose of this study was to evaluate myocardial dysfunction in cats with SIRS via cardiac troponin and ECG findings.” Or similar.
- Done.
Line 17 – the results of this study demonstrate cats with SIRS have increased cardiac troponin levels and ECG alterations, consistent with myocardial dysfunction.
- Done.
Abstract
- Line 19 – recommend “human studies have demonstrated…” Again, does it matter?
- Modified.
- Line 20 – recommend rewording – this study evaluated the concentration of Troponin I and ECG findings…
- Done.
- Line 23 – while 10 healthy cats were evaluated as a control population\
- Done.
- Line 27 – replace “breath” with respiratory.
- Done.
- Overall it would be more helpful to separate how cats were included – criteria met and then talk about what diagnostics were performed in all cats once they were enrolled – ie: troponin and ECG
Define healthy cats? How were they enrolled?
More information about ECG in both groups should be provided.
- All information required have been included.
Introduction
- Line 37 – citations all at the end of the sentence
- Done.
- Line 40 – Indent for new paragraph.
- Done.
- Line 42-44 – this sentence doesn’t seem to add much to the point of the intro.
- Modified.
- Line 46 – “prevention” should likely not be in this sentence since it’s not the point of the study.
- Modified.
- Line 47 – “presence of at least 4 criteria” reads confusing because they don’t need to meet all criteria.
- Modified.
- Line 50 – cats and dogs need to meet a different # of criteria, so this should be reworded.
- Modified.
- Line 51 “numerous studies...” this is not relevant to the intro and should not be here.
- Modified.
- Line 54 – this should be reordered – discuss the effects of SIRS on the body (MODS) and then more specifically on cardiac effects and prognosis.
- Done.
- Line 61 – this paragraph needs to be reorganized/rewritten; consider “Troponin I is specific for myocardial damage and elevation of troponin I is reported in…”.
- Modified.
- Line 68 – remove this sentence.
- Done.
- Line 69 – citation?
- Done.
- Line 69- I am not sure this is true; “consistently supposing believed” is grammatically incorrect.
- Modified.
- Line 72 – reword “the goal of this study”
- Done.
- Line 75 – remove this sentence.
- Done.
Materials/Methods
- Line 82 – during hospitalization should not be criteria – what time frame? What time frame was it diagnosed with data collected.
- All information required have been included.
- Line 87 – why were these excluded? Based on what?
- All information required have been included.
- Line 87 – history of cardiac disease based on what? History of kidney disease based on what?\
- All information required have been included.
- Line 88 – this is unclear and makes it seem biased – not all cats had an echo and some were excluded based on the echo?
- The paragraph has been modified.
- Line 90 – how were they recruited? Did they also need to weight > 2kg and between 6 months to 12 years?
- Given that other factors may be influence the cTnI concentration, as reported by the references.
- Line 103 – at what time frame?
- The information has been included.
- Line 125 – what time frame?
- The information has been included.
- Line 133-163 – too much information.
- It needs to include discussing of ECG examination.
- Line 133 – evaluated by who? A cardiologist?
- Of course, it is evaluated by a cardiologist (M.P.).
Results
- Line 176 – what does this mean?
- The sentence has been modified.
- Line 177 – this should not be at the beginning of this paragraph
- Done.
- Line 182 – information on which criteria they met should be provided – each cat needed to meet 2/4 – which 2 did they meet.
- Modified.
- Line 192 – doesn’t seem like information that needs to be presented; they have to have significantly different scores in order to meet inclusion criteria.
- Modified.
- Line 195 – is this supposed to be bradycardia?
- Modified.
R, Line 194 – confusing statement; state how many were tachycardic and how many were bradycardic.
- Modified.
- Line 195 – needs grammatical correction.
- Done.
- Line 192-202 – information should be provided for both groups and statistical difference provided for all values; not just MAP and troponin.
- Modified.
- No information was even provided about BP (in text and table).
- Included.
- Line 204 – HR is good to note; do other values matter? How often are these really looked at?
- We have evaluated criteria of SIRS and ECG. Other evaluation performed during the monitoring of patients are not included in the aim of this study.
- Line 210 – what about group 2?
- All information required have been included.
- Line 210 – what does this mean?
- Modified.
- What cats had echos? Why did they have echos? It is mentioned earlier that some cats had echos and were excluded but there is no mention in the results section.
- Modified.
Discussion
- Line 217 – what do you mean significant?
- Modified.
- Line 218 – “following…” should be removed.
- Done.
- Line 219 – doesn’t seem to contribute to the discussion
- Modified.
- Line 221 – this sentence is not grammatically correct; also not necessarily true – diagnosis is the first important step.
- Modified.
- Line 222 – need grammatical corrections.
- Done.
- Line 224 – is this relevant to the study?
- Modified.
- Line 230 – run on sentence; needs to be adjusted
- Modified.
- Line 230 – this is a disjointed paragraph.
- Modified.
- Line 235 – troponin is not being evaluated as a biomarkers for SIRS.
- Of course. This is considered a biomarker of cardiac dysfunction in the course of SIRS.
- Line 235-240 – this doesn’t make sense because this study is not looking at biomarkers for diagnosing SIRS.
- Modified.
- Line 246 – involve, not involves
- Done.
- Line 249 – it’s unclear what the point of these statements are
- Modified,
- The discussion is very unorganized and hard to follow – based on this discussion it is hard to know what the points are being made.
- The discussion has been revised and improved.
- Unsure the conclusions and how they are aid in diagnosis or management.
- Modified.
Round 2
Reviewer 1 Report
I thank the authors for their attention and responses to previous comments, however it is my believe that the manuscript as a whole still requires an intensive revision of the English language Also some additional minor comments and doubts: Line 79 – define “cTnI” lines 118/23 - once again methodology concerning blood pressure measurements should be more detailed. Information is missing regarding to cover letter Line 129 - Please use “eletrolytes” instead of “ions” as in abbstract Line 133 - Please use “feline white blood cell morphology” instead of “feline white cell morphology” Line 199/200 - Please consider to move this sentence to after the paragraph “Group 1 was composed of 17 domestic shorthair cats (10 males and 7 females, aged 188 1.3–11.2 years (median age 6.2 years, IQR 8.8 years) and weighing 2.2–6.1 kg (average 189 3.75).“ Line 207 - anterior version refer to 4 cats with bradypnea, know authors refer to 4 cats with bradycardia… Lines 211/4 - cats with high temperature or leucocytosis shouldn't be considered normal … please consult the M&M section
Table 3 - “MEA” - All the abbreviations used in the table should be defined in foot notes
It is my believe that the manuscript as a whole still requires an intensive revision of the English language
Author Response
Dear Reviewer,
Thank you very much for your time and all your comments.
We thank for precise and thoughtful comments and constructive criticism, which has led to a better manuscript.
We revised the manuscript in relation to the suggestions and more detailed answers are given below.
The changes made in the manuscript to address comments are written in red.
As you are suggested a further English language revision has been performed.
- thank the authors for their attention and responses to previous comments, however it is my believe that the manuscript as a whole still requires an intensive revision of the English language.
- As you are suggested a further English language revision has been performed.
- Line 79 – define “cTnI” lines 118/23.
- A. Done.
- once again methodology concerning blood pressure measurements should be more detailed. Information is missing regarding to cover letter.
- The methodology concerning blood pressure measurements are better detailed.
- Line 129 - Please use “eletrolytes” instead of “ions” as in abstract.
- Done.
- Line 133 - Please use “feline white blood cell morphology” instead of “feline white cell morphology” .
- Done.
- Line 199/200 - Please consider to move this sentence to after the paragraph “Group 1 was composed of 17 domestic shorthair cats (10 males and 7 females, aged 188 1.3–11.2 years (median age 6.2 years, IQR 8.8 years) and weighing 2.2–6.1 kg (average 189 3.75).
- Done.
- Line 207 - anterior version refer to 4 cats with bradypnea, know authors refer to 4 cats with bradycardia…
- I’m sorry, a mistake was present.
- Lines 211/4 - cats with high temperature or leucocytosis shouldn't be considered normal … please consult the M&M section.
- As reported in M&M cats with at least two of the four following parameters such as abnormal body temperature, abnormal heart rate, abnormal breath rate and white blood cells (WBC) abnormalities (WBC ≥19500 or ≤5000 k/μL or band neutrophils ≥5%) were considered affected with SIRS.
- Table 3 - “MEA” - All the abbreviations used in the table should be defined in foot notes.
- Done.
Best regards
Prof. Michela Pugliese